# Dietary Nitrate Intake Is Associated with Decreased Incidence of Open-Angle Glaucoma: The Rotterdam Study

**DOI:** 10.3390/nu14122490

**Published:** 2022-06-15

**Authors:** Joëlle E. Vergroesen, Tosca O. E. de Crom, Lauren C. Blekkenhorst, Caroline C. W. Klaver, Trudy Voortman, Wishal D. Ramdas

**Affiliations:** 1Department of Ophthalmology, Erasmus MC University Medical Center, P.O. Box 2040, 3000 CA Rotterdam, The Netherlands; j.vergroesen@erasmusmc.nl (J.E.V.); c.c.w.klaver@erasmusmc.nl (C.C.W.K.); 2Department of Epidemiology, Erasmus MC University Medical Center, P.O. Box 2040, 3000 CA Rotterdam, The Netherlands; t.decrom@erasmusmc.nl (T.O.E.d.C.); trudy.voortman@erasmusmc.nl (T.V.); 3School of Medical and Health Sciences, Edith Cowan University, Joondalup, WA 6027, Australia; l.blekkenhorst@ecu.edu.au; 4Department of Ophthalmology, Radboud University Medical Center, P.O. Box 9101, 6500 HB Nijmegen, The Netherlands; 5Institute of Molecular and Clinical Ophthalmology, University of Basel, CH-4031 Basel, Switzerland; 6Division of Human Nutrition and Health, Wageningen University & Research, P.O. Box 17, 6700 AA Wageningen, The Netherlands

**Keywords:** open-angle glaucoma, intraocular pressure, dietary nitrate, green-leafy vegetables

## Abstract

Previous studies suggest that nitric oxide is involved in the regulation of the intraocular pressure (IOP) and in the pathophysiology of open-angle glaucoma (OAG). However, prospective studies investigating the association between dietary nitrate intake, a source of nitric oxide, and incident (i)OAG risk are limited. We aimed to determine the association between dietary nitrate intake and iOAG, and to evaluate the association between dietary nitrate intake and IOP. From 1991 onwards, participants were followed each five years for iOAG in the Rotterdam Study. A total of 173 participants developed iOAG during follow-up. Cases and controls were matched on age (mean ± standard deviation: 65.7 ± 6.9) and sex (%female: 53.2) in a case:control ratio of 1:5. After adjustment for potential confounders, total dietary nitrate intake was associated with a lower iOAG risk (odds ratio (OR) with corresponding 95% confidence interval (95% CI): 0.95 (0.91–0.98) for each 10 mg/day higher intake). Both nitrate intake from vegetables (OR (95% CI): 0.95 (0.91–0.98) for each 10 mg/day higher intake) and nitrate intake from non-vegetable food sources (OR (95% CI): 0.63 (0.41–0.96) for each 10 mg/day higher intake) were associated with a lower iOAG risk. Dietary nitrate intake was not associated with IOP. In conclusion, dietary nitrate intake was associated with a reduced risk of iOAG. IOP-independent mechanisms may underlie the association with OAG.

## 1. Introduction

Glaucoma is an eye disease that causes the most cases of irreversible blindness worldwide. Currently, more than 80 million people worldwide have glaucoma, of which approximately 11 million are estimated to be bilaterally blind [1]. A high intraocular pressure (IOP) is a well-known modifiable risk factor but, since glaucoma can progress despite an “adequate” IOP, it is very likely that IOP-independent mechanisms play a role as well. Therefore, more knowledge about other potential risk factors is urgently needed for optimal prevention and treatment strategies.

Several studies have investigated the association between nutrition and open-angle glaucoma (OAG) [2]. Studies on the intake of dark green leafy vegetables showed an inverse association with OAG [3,4,5]. This may in part be explained by the substantial amount of dietary nitrate that green leafy vegetables contain (2000–5000 mg/kg) [6,7,8], along with phylloquinone, lutein, folate, α-tocopherol, and kaempferol [9]. Due to the nitrate–nitrite–nitric oxide pathway, nitrate is an important source of nitric oxide (NO). Different studies have suggested that NO, known as endothelium-derived relaxing factor [10], plays a role in the regulation of IOP, by increasing the conventional outflow facility [3,11,12,13,14,15]. Abnormal function and degradation of endothelial cells are associated with reduced NO bioavailability and, subsequently, progression of glaucoma [16]. Additionally, the endothelium of Schlemm’s canal (SC) reacts to physiological levels of shear stress, by aligning with the direction of flow and by increasing the production of NO. NO production by SC cells has a homeostatic signaling function during times of elevated IOP, when SC narrows and shear stress on SC cells increases. Shear-stimulated production of NO by SC cells would then increase outflow facility, normalizing IOP [17,18,19,20]. This process may be compromised in glaucoma, as SC cells isolated from glaucomatous eyes have shown to be either shear-unresponsive or lifted from their substrate in the presence of shear stress [21]. IOP-independent effects of dietary nitrate have also been suggested. Dietary nitrate has shown to have beneficial effects on blood pressure, endothelial function, reperfusion injury, and platelet aggregation [22]. All of these may be involved in the pathophysiology of OAG, but studies investigating whether dietary nitrate intake relates to the risk of incident (i) OAG are limited.

The aim of this study was to determine the association between dietary nitrate intake and iOAG. We also examined the association between dietary nitrate intake and IOP, as an OAG risk factor, and we studied whether potential associations with iOAG were explained by IOP or, indirectly, by blood pressure.

## 2. Materials and Methods

### 2.1. Study Design and Population

Participants were derived from three independent cohorts from the prospective population-based Rotterdam Study (RS-I, RS-II, RS-III), designed to assess determinants of age-related diseases in the middle-aged and elderly population (45+ years). Enrollment for the ophthalmic part started in 1991; after the baseline visit, participants were invited for follow-up visits with intervals of approximately five years [23]. Of 8679 participants with ophthalmic examinations, 7008 had baseline measurements of dietary nitrate intake. Of those, 173 participants developed iOAG during follow-up. Since age is strongly associated with iOAG risk [24] and dietary intake [25,26], and dietary intake is different for females compared to males [27], we chose to use a case–control design. We matched cases and controls on age (maximum difference of three years) and sex, in a 1:5 ratio, and sampled without replacement. The final dataset consisted of 173 cases and 865 controls.

### 2.2. Ophthalmic Assessment

The eye examinations included Goldmann applanation tonometry (Haag-Streit AG, Bern, Switzerland), and visual field testing (Humphrey Field Analyzer; HFA II 740; Carl Zeiss, Oberkochen, Germany). All participants underwent visual field testing using the Humphrey Field Analyzer (HFA; Carl Zeiss Meditec, Jena, Germany). A second supra-threshold test was performed when a visual field defect appeared to be present. Details have been described elsewhere [28]. If the second supra-threshold test showed at least one overlapping abnormality in the same hemifield, Goldmann kinetic perimetry (RS-I-1 and RS-I-3; Haag-Streit) or full-threshold HFA (all other cohort visits) was performed on both eyes. If abnormalities were consecutive and reproducible, thus present on the Goldmann or full-threshold test and on both supra-threshold tests, visual field loss was considered to be present. Defects had to be in a consistent hemifield and at least one depressed test point had to have exactly the same location on all fields. Glaucoma specialists examined fundus photographs, ophthalmic examination reports, medical histories, and MRI scans of the brain to exclude all other possible causes of visual field loss. Discrepancies were resolved by consensus. iOAG cases had an open anterior chamber angle and no history or signs of secondary glaucoma [28]. For IOP, three measurements were taken from each eye, the median value of which was recorded [29]. For iOAG cases, we used IOP measurements of the affected eye. If both eyes were affected or unaffected, a random eye was selected. IOP was not included in the definition of iOAG.

### 2.3. Dietary Nitrate Data

Dietary intake was assessed at baseline using food frequency questionnaires (FFQs) as described in detail elsewhere [30]. Both FFQs were previously validated and showed reasonable to good estimates of nutrient intake [31,32,33]. All food items were assessed based on the frequency of consumption, the number of servings per day as well as on the preparation methods. We calculated dietary nitrate intake separately from vegetables and non-vegetable food sources, because of their possible contradicting health effects [34,35,36,37,38,39]. Nitrate intake for each vegetable was calculated using a comprehensive database, including nitrate data for 178 vegetables from over 250 publications [40]. Nitrate intake from vegetables (mg/day) was calculated by multiplying the amount of each vegetable (g/day) by the median nitrate content (mg/g) for that individual vegetable. Nitrate intake from non-vegetable food sources was obtained from an earlier developed dietary nitrate and nitrate database [41]. Nitrate intake from non-vegetable food sources was estimated by multiplying the amount of the food item (g/day) by the mean nitrate value (mg/g) of that food item. If no nitrate value was available for a specific food item, we considered a value of 0 mg/g. Total dietary nitrate intake (mg/day) was calculated by summing the nitrate intake from vegetables and nitrate intake from non-vegetable food sources. Participants with unreliable dietary intake (total energy intake <500 kcal/day or >5000 kcal/day) were excluded.

### 2.4. Covariates

Education level was assessed with questionnaires and categorized into: primary education, lower education, intermediate education, or higher education. Smoking status was obtained using questionnaires and participants were classified as non-smoker, former smoker or current smoker. At the research center, blood pressure was measured at the right brachial artery with the participant in sitting position. The mean of two consecutive measurements was used. Hypertension was defined as a resting blood pressure exceeding 140/90 mmHg or the use of blood pressure-lowering medication. Medication data on blood pressure-lowering medications (antihypertensives, diuretics, beta blockers, calcium channel blockers, and renin-angiotensin-aldosterone system agents) were collected with questionnaires [42]. Weight and height were measured at the research center. Body mass index (BMI) was calculated as weight in kilograms divided by height in meters squared. Total energy intake was obtained from the previously described FFQs. Diet quality was defined as adherence to the Dutch dietary guidelines, with a scoring range from 0 (no adherence) to 14 (full adherence). Details have been described elsewhere [43]. For physical activity, two different questionnaires were used: a validated adapted version of the Zutphen Physical Activity Questionnaire [44] and the LASA Physical Activity Questionnaire [45]. Data were recalculated into metabolic equivalent of task (MET)-hours per week, and a z-standardized score was included in the analyses.

### 2.5. Statistical Analyses

Differences in baseline characteristics between cases and controls were evaluated using chi-square tests and independent-samples t-tests. We adjusted dietary nitrate intake for total energy intake by applying the nutrient residual method and analyzing the dietary nitrate intake adjusted for total energy intake. One-way ANOVA was used to compare the baseline characteristics of participants in the different quintiles of total dietary nitrate intake. The dose–response relationship between dietary nitrate intake and predicted iOAG probability or IOP, was examined using generalized additive modelling. We performed multivariable conditional logistic regression analyses to calculate odds ratios (ORs) with corresponding 95% confidence intervals (CI) for iOAG and hypertension. ORs can be interpreted as the difference in odds per increase of 10 mg/day intake of dietary nitrate keeping energy intake constant (iso-energetic). Additionally, we modelled dietary nitrate intake in quintiles with the first quintile (Q1) as reference category to test for evidence of linear trends. The median value for each category as continuous variables was used in separate conditional logistic regression models. The final models included BMI, total energy intake, diet quality, physical activity, and follow-up time. Follow-up duration was calculated from the baseline until the last visit with reliable ophthalmic examination or the first visit with iOAG diagnosis. To assess potential reverse causality, we analyzed the association between dietary nitrate intake and iOAG in cumulative follow-up intervals. Additionally, we observed the effect of including IOP (potential mediator in the association with iOAG) or education level and smoking status (lifestyle factors affecting nutrition quality) in the models. The association of dietary nitrate with IOP at follow-up, and diastolic and systolic blood pressure at baseline, was assessed by performing multivariable linear regression analysis, adjusting for the same covariates as mentioned above. The blood pressure analyses were additionally adjusted for use of blood pressure-lowering medications. Statistical analyses were performed using SPSS v25.0 (SPSS Inc., Chicago, IL, USA) and R v3.6.1 (R Inc., Boston, MA, USA), with packages DescTools, mgcv, ggplot2, dplyr and ggforestplot. A *p*-value < 0.05 was considered statistically significant.

## 3. Results

The baseline characteristics of cases and controls are displayed in Table 1. Participants with iOAG had a significantly lower BMI and their diet quality score was higher. As expected, they had a significantly higher IOP. Dietary nitrate intake was significantly different between cases and controls. Baseline characteristics according to quintiles of total dietary nitrate intake are presented in Table 2. Higher consumers of dietary nitrate more often had a higher education. Additionally, their BMI, total energy intake and diet quality score were higher.

Figure 1 presents a graphic representation of the dose–response relationship between dietary nitrate intake, iOAG and IOP analyzed in separate generalized additive multivariable-adjusted models. For iOAG, similar dose–response relationships were found for total dietary nitrate intake (Figure 1A), nitrate intake from vegetables (Figure 1B) and nitrate intake from non-vegetable food sources (Figure 1C), i.e., they were linear across the reported range of intake. For IOP, a different dose–response relationship was found for nitrate intake from non-vegetable food sources and IOP (Figure 1F) compared to the relationship with total dietary nitrate intake and nitrate intake from vegetables (Figure 1D,E). The association of nitrate intake from non-vegetable food sources with IOP was linear across the reported range of intake, whereas the associations of total dietary nitrate intake and nitrate intake from vegetables with IOP were not.

In the multivariable-adjusted model (Figure 2A; model 1), each 10 mg/day higher total dietary nitrate intake was associated with a 5% reduction in the risk of iOAG (OR (95% CI): 0.95 (0.91–0.98)). Participants in the highest quintile (Q5: mean 213.0 mg/day) had the largest risk reduction (OR (95% CI): 0.38 (0.20–0.72)) compared to participants in the lowest quintile (Q1: mean 48.8 mg/day) (*p*-trend = 0.002). For nitrate intake from vegetables, we observed a 5% reduction in the risk of iOAG (OR (95% CI): 0.95 (0.91–0.98)) for each 10 mg/day higher intake (Figure 2B; model 1). The difference in iOAG risk was 61% when comparing the highest (Q5: mean 196.8 mg/day) and lowest (Q1: mean 34.6 mg/day) nitrate intake from vegetables (OR (95% CI): 0.39 (0.20–0.73)) (*p*-trend = 0.003). For nitrate intake from non-vegetable food sources, we observed a 37% reduction in the risk of iOAG (OR (95% CI): 0.63 (0.41–0.96)) for each 10 mg/day higher intake (Figure 2C; model 1), but we did not observe a significant trend (*p*-trend = 0.08). Additional adjustment of the aforementioned analyses with IOP (Figure 2; model 2) or with education level and smoking status (Figure 2; model 3) did not change the results. When analyzing the cumulative follow-up intervals, a higher intake of dietary nitrate intake was associated with a lower iOAG risk during every cumulative follow-up interval after 10 years of follow-up (Appendix A).

For IOP as outcome, we observed no significant associations with total dietary nitrate intake (beta (95% CI): 0.02 (−0.02–0.06) for each 10 mg/day higher intake) and nitrate intake from vegetables (beta (95% CI): 0.02 (−0.02–0.06) for each 10 mg/day higher intake) (Table 3). We did observe a borderline significant association between nitrate intake from non-vegetable food sources and IOP (beta (95% CI): −0.45 (−0.96–0.06) for each 10 mg/day higher intake) (*p*-trend = 0.09). We found no significant associations between dietary nitrate intake and diastolic blood pressure (Appendix A) and systolic blood pressure (Appendix A). Only nitrate intake from non-vegetable food sources was associated with a lower risk of hypertension (OR (95% CI): 0.65 (0.45–0.94) for each 10 mg/day higher intake) (*p*-trend = 0.06) (Appendix A).

## 4. Discussion

In this case–control study embedded within a prospective population-based cohort, we found that dietary nitrate intake showed a strong association with a decreased incidence of OAG. No significant associations were observed between dietary nitrate intake and IOP. Additionally, no clear associations were observed between dietary nitrate intake and blood pressure.

To our knowledge, we are the first to assess the association between dietary nitrate and iOAG, stratified by source (vegetables vs. non-vegetable food sources). The Nurses’ Health Study and the Health Professionals Follow-up Study reported a pooled multivariable rate ratio (MVRR) of 0.79 (95% CI 0.66–0.93; *p*-trend = 0.02) for the highest quintile of dietary nitrate intake (~240 mg/day) as compared with the lowest quintile (~80 mg/day) [3]. When additionally adjusted for other dietary factors, this pooled MVRR decreased to 0.67 (95% CI 0.52–0.85; *p*-trend = 0.01) [3]. We found a similar result and trend. A nitrate intake of ~200 mg can be achieved by consuming 100 g spinach (nitrate: 1926 mg/kg), 130 g beets (nitrate: 1581 mg/kg), 190 g endive (nitrate: 1054 mg/kg) or 115 g kale (nitrate: 1748 mg/kg) [40]. These are very feasible portion sizes, as the Dutch dietary guidelines recommend consuming at least 200 g of vegetables daily [43]. As we did not observe an association between dietary nitrate intake and IOP, the association between dietary nitrate and iOAG may be explained by other, IOP-independent, mechanisms.

Dietary nitrate intake may affect the risk of iOAG due to its beneficial effects on blood pressure, endothelial function, reperfusion injury, and platelet aggregation (Appendix A). These effects are likely to occur as a result of enhanced NO production through the nitrate–nitrite–NO pathway [22]. Previous research has shown that a higher dietary nitrate intake was associated with significantly wider retinal arterioles [46]. Widening of retinal arteriolar caliber is not only associated with a lower risk of cardiovascular and cerebrovascular diseases [47,48], but also with a lower risk of glaucoma [49]. Eyes with primary OAG were 2.7 times more likely to have generalized arteriolar narrowing, and narrower retinal arterioles were significantly associated with higher OAG prevalence and incidence [50,51,52,53,54]. Thus, the association between nitrate and OAG may be explained by increased retinal arteriolar caliber caused by nitrate, which affects blood pressure. Previous population-based studies have suggested that IOP is associated with systemic blood pressure levels [55,56,57,58,59,60,61,62,63,64]. Nevertheless, in our study, we found no clear association between dietary nitrate intake and IOP or blood pressure.

Although research into the IOP-independent pathways by which dietary nitrate intake could influence glaucoma incidence are limited, we would like to highlight their potential in explaining in part or in combination the inverse association found in this study. Endothelial dysfunction in glaucoma has been associated with an imbalance between endothelin-1 and NO [65]. Dietary nitrate could thus potentially lower the incidence of OAG by upregulating the NO levels, hereby improving endothelial function. Glaucomatous retinal ganglion cell loss has previously been associated with increased oxidant levels [66,67,68,69], a theory that is supported by the fact that administration of antioxidants protects retinal cells from injury following retinal ischemia and reperfusion [70,71,72,73]. Retinal ischemia can thus potentially impact optic nerve degeneration [74]. Increased NO bioavailability acts on the balance between antioxidants and prooxidant agents [75]. NO can eliminate oxidants, reduce equivalents provided by superoxide, and prevent the reaction of peroxide [76]. Dietary nitrate has shown to suppress radical formation and to be a scavenger of potentially damaging reactive oxygen and nitrogen species, suggesting that it may also exhibit antioxidant effects [76,77]. This is one mechanism that may play a role in the observed association between dietary nitrate intake and iOAG. Moreover, adhesion and aggregation of platelets is inhibited by NO. Modulation of platelet function is an important therapeutic strategy in preventing and treating atherosclerosis, a disease considered to increase glaucoma risk [78,79]. Thus, mediation of platelet aggregation is one other mechanism that could underlie the association between dietary nitrate intake and iOAG.

This study has several strengths. We used a prospective population-based design, allowing repeated eye examinations, and thus prospectively ascertaining iOAG cases, according to a well-established OAG definition [28] and IOP measurements. Additionally, dietary data were collected using validated FFQs, which included a wide variety of food items commonly consumed in the Dutch population. By using dietary information from baseline assessments, we limited selection bias and the risk of reverse causality, since all included participants were free of iOAG at this visit. Moreover, the questionnaire was administered to cases and controls under similar conditions. Furthermore, we assessed the association between dietary nitrate intake and iOAG over cumulative follow-up periods to provide insight into possible reversed causality. The persistence of the association over time implies that reverse causality is unlikely. The availability of robust data on possible confounders allowed us to reach an independent association between dietary nitrate intake and iOAG. Given that our cases and controls were matched on age and sex, it is very unlikely that our findings were affected by the association of age and sex with dietary (nitrate) intake. We performed additional matching on BMI (with a range of 2.0 kg/m^2^), since the controls in this study had a significantly higher BMI than the iOAG cases, and a higher BMI appears to be associated with lower iOAG risk [80,81,82,83,84,85]. However, additional matching on BMI did not change the association between dietary nitrate intake and iOAG or IOP (Appendix A, respectively). Limitations should also be considered when interpreting our results. By assessing the association in time, thus only looking at incident disease, we limited the number of iOAG cases, and therefore also IOP measurements. As the iOAG cases did not have exorbitant IOP measurements typically associated with OAG (mean 16.2 mmHg; interquartile range 13–18 mmHg), this may have limited our possibilities to detect statistically significant IOP-lowering effects of dietary nitrate intake. By using the FFQ, we relied on the participants’ memory for collecting information for as far back as one month. Additionally, the FFQ is known to under- or over-report certain foods, leading to non-differential misclassification. Additionally, based on the FFQs, nitrate intake over the past year or month was determined, which does not per definition reflect long term intake as participants may change dietary habits over time. However, since dietary information was collected at baseline, with all participants free of iOAG, it is unlikely that such misclassification would result in false-positive findings. If glaucoma presence would have an effect on dietary nitrate intake, this would not be applicable to our study. Despite the limitations, the low respondent burden makes the FFQ an easy and effective data collection tool. It additionally allows for calculation of the total energy intake, which is a large benefit [86]. Although the analyses were adjusted for multiple confounders, we were unable to adjust for other possible confounders such as family history of glaucoma, since this was only available for a small subset of participants. We did consider the risk factor myopia, for which we adjusted by including education level into model 3. We also included spherical equivalent into the model (data not shown), but this did not change the results. Lastly, residual confounding cannot completely be excluded. In summary, a higher dietary nitrate intake reduces the risk of iOAG. The effect was independent of the IOP. Our findings confirm earlier reported associations between dietary nitrate intake and OAG. However, intervention studies are necessary before the association between dietary nitrate intake and iOAG can be considered as an important public health implication.

## Figures and Tables

**Figure 1 nutrients-14-02490-f001:**
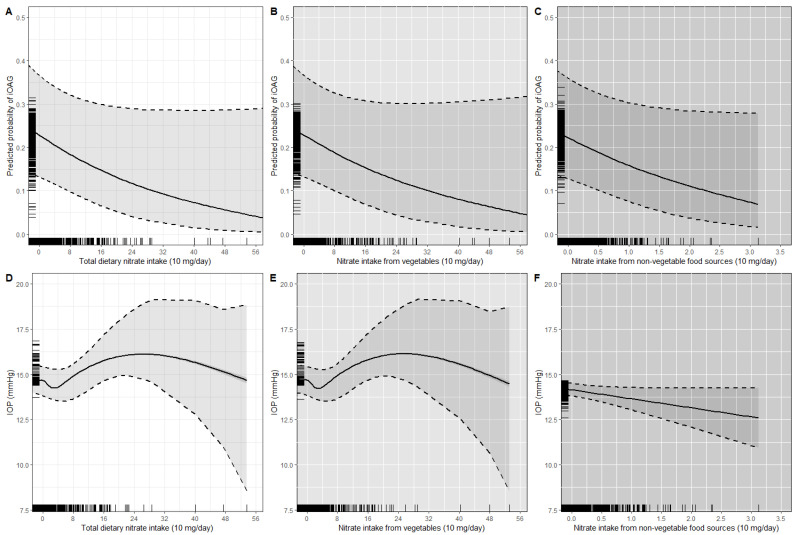
Graphic presentation of the multivariable-adjusted dose–response relationship between incident open-angle glaucoma (iOAG), intraocular pressure (IOP), and energy adjusted dietary nitrate intake obtained by generalized additive regression models; total dietary nitrate intake (**A**,**D**), nitrate intake from vegetables (**B**,**E**), and nitrate intake from non-vegetable food sources (**C**,**F**). Dotted lines represent 95% confidence intervals. The reference value is the value associated with the mean nitrate intake for all participants. The rug plot along the *x*- and *y*-axis of each graph depicts each observation.

**Figure 2 nutrients-14-02490-f002:**
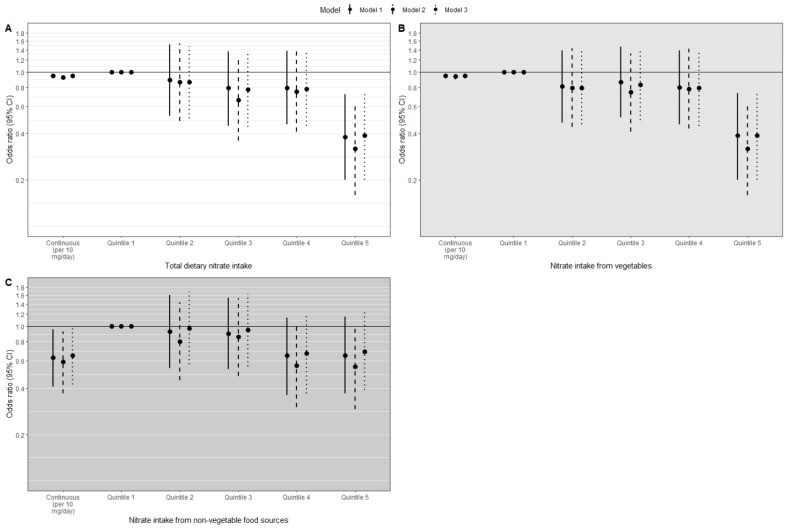
Odds ratios (95% confidence interval (CI)) for open-angle glaucoma by total dietary nitrate intake (**A**), nitrate intake from vegetables (**B**), and nitrate intake from non-vegetable food sources (**C**) (as continuous variables and quintiles) analyzed using conditional logistic regression. Model 1: adjusted for body mass index, total energy intake, diet quality, physical activity, and follow-up time. Model 2: model 1 additionally adjusted for intraocular pressure. Model 3: model 1 additionally adjusted for education level and smoking status.

**Table 1 nutrients-14-02490-t001:** Baseline characteristics of participants that did and did not develop incident open-angle glaucoma (iOAG) during follow-up.

	No iOAG (N = 865)	iOAG (N = 173)	*p*-Value
Age, years, mean (SD)	64.8 (7.0)	65.7 (6.9)	0.12
Sex, female, N (%)	460 (53.2)	92 (53.2)	>0.99
Education, N (%)			0.77
Primary education	101 (11.7)	21 (12.1)	
Lower education	376 (43.5)	78 (45.1)	
Intermediate education	250 (28.9)	53 (30.6)	
Higher education	131 (15.1)	21 (12.1)	
Smoking status, N (%)			0.79
Non-smoker	281 (32.5)	54 (31.2)	
Former smoker	410 (47.4)	81 (46.8)	
Current smoker	170 (19.7)	38 (21.9)	
Hypertension, N (%)	491 (56.8)	92 (53.2)	0.46
SBP, mmHg, mean (SD)	137.6 (20.6)	136.6 (20.9)	0.58
DBP, mmHg, mean (SD)	77.1 (11.6)	75.0 (12.5)	0.03
BMI, kg/m2, mean (SD)	27.1 (4.1)	25.9 (3.3)	<0.001
Total energy intake, kcal/day, mean (SD)	2119.1 (594.5)	2054.3 (515.0)	0.19
Diet quality, mean (SD)	6.6 (1.9)	7.0 (1.9)	0.04
Physical activity, MET hours/week, mean (SD)	0.0 (0.9)	0.1 (0.9)	0.07
IOP, mmHg, mean (SD)	14.1 (2.9)	16.4 (3.9)	<0.001
Follow-up time, years, mean (SD)	9.5 (4.7)	10.9 (5.3)	<0.001
Total dietary nitrate intake, mg/day, mean (SD)	109.8 (78.4)	92.8 (47.1)	<0.001
Nitrate intake from vegetables, mg/day, mean (SD)	94.2 (76.3)	77.4 (45.2)	<0.001
Nitrate intake from non-vegetable food sources, mg/day, mean (SD)	15.6 (7.9)	15.4 (10.9)	0.78

Abbreviations: iOAG, incident open-angle glaucoma; N, number; SBP, systolic blood pressure; DBP, diastolic blood pressure; BMI, body mass index; MET, metabolic equivalent of task; IOP, intraocular pressure; SD, standard deviation.

**Table 2 nutrients-14-02490-t002:** Baseline characteristics of participants by energy adjusted total dietary nitrate intake (1st, 3rd, and 5th quintiles).

KERRYPNX	Q1 (N = 205)	Q3 (N = 206)	Q5 (N = 205)	*p* ANOVA
iOAG, N (%)	38 (18.5)	38 (18.4)	20 (9.8)	0.07
Age, years, mean (SD)	66.4 (7.1)	65.7 (6.9)	62.5 (6.0)	<0.001
Sex, female, N (%)	93 (45.4)	124 (60.2)	119 (58.0)	0.03
Education, N (%)				0.005
Primary education	31 (15.1)	29 (14.1)	7 (3.4)	
Lower education	86 (42.0)	89 (43.2)	94 (45.8)	
Intermediate education	61 (30.0)	58 (28.2)	60 (29.3)	
Higher education	24 (11.7)	28 (13.6)	43 (21.0)	
Smoking status, N (%)				0.73
Non-smoker	69 (33.7)	75 (36.4)	60 (29.3)	
Former smoker	88 (42.9)	91 (44.2)	106 (51.7)	
Current smoker	46 (22.4)	40 (19.4)	39 (19.0)	
Hypertension, N (%)	118 (57.5)	123 (59.7)	118 (57.6)	0.39
SBP, mmHg, mean (SD)	140.8 (20.4)	138.9 (22.9)	137.0 (19.7)	0.04
DBP, mmHg, mean (SD)	77.7 (12.3)	76.5 (12.5)	78.4 (10.6)	0.02
BMI, kg/m2, mean (SD)	26.3 (3.6)	26.8 (3.9)	28.2 (4.6)	<0.001
Total energy intake, kcal/days, mean (SD)	2233.2 (673.9)	2024.8 (484.4)	2140.5 (569.8)	0.002
Diet quality, mean (SD)	6.0 (1.8)	7.2 (1.8)	7.0 (2.0)	<0.001
Physical activity, MET hours/week, mean (SD)	−0.1 (0.9)	0.1 (0.9)	0.1 (0.9)	0.06
IOP, mmHg, mean (SD)	14.6 (3.1)	14.6 (3.1)	14.1 (3.4)	0.33
Follow-up time, years, mean (SD)	9.6 (4.5)	10.4 (5.2)	9.2 (4.8)	0.07
Total dietary nitrate intake, mg/day, mean (SD)	48.8 (15.7)	86.4 (11.4)	213.0 (91.7)	<0.001
Nitrate intake from vegetables, mg/day, mean (SD)	35.1 (14.5)	71.2 (10.1)	196.8 (91.4)	<0.001
Nitrate intake from non-vegetable food sources, mg/day, mean (SD)	13.7 (4.6)	15.2 (5.7)	16.3 (6.7)	<0.001

Abbreviations: N, number; Q, quintile; ANOVA, analysis of variance; iOAG, incident open-angle glaucoma; SBP, systolic blood pressure; DBP, diastolic blood pressure; BMI, body mass index; MET, metabolic equivalent of task; IOP, intraocular pressure; SD, standard deviation.

**Table 3 nutrients-14-02490-t003:** Multivariable adjusted beta (95% confidence interval) of intraocular pressure (IOP), by quintiles of nitrate intake.

		Beta ^a^ per 1 Unit Increase	*p*-Value	Q1	Q2	Q3	Q4	Q5	*p*-Trend ^b^
Total dietary nitrate intake (10 mg/day)	Model 1	0.02 (−0.02–0.06)	0.35	0.00	−0.04 (−0.89–0.80)	−0.25 (−1.05–0.55)	−0.22 (−0.98–0.53)	−0.15 (−0.99–0.69)	0.78
	Model 2	0.02 (−0.02–0.06)	0.39	0.00	−0.02 (−0.87–0.83)	−0.30 (−1.11–0.50)	−0.23 (−0.99–0.54)	−0.20 (−1.06–0.66)	0.69
Nitrate intake from vegetables (10 mg/day)	Model 1	0.02 (−0.02–0.06)	0.29	0.00	0.33 (−0.48–1.13)	0.17 (−0.61–0.95)	−0.13 (−0.85–0.60)	0.11 (−0.69–0.91)	0.91
	Model 2	0.02 (−0.02–0.06)	0.32	0.00	0.33 (−0.48–1.14)	0.18 (−0.61–0.97)	−0.12 (−0.85–0.62)	0.05 (−0.76–0.87)	0.82
Nitrate intake from non-vegetable food sources (10 mg/day)	Model 1	−0.45 (−0.96–0.06)	0.09	0.00	0.37 (−0.52–1.25)	0.05 (−0.73–0.84)	−0.15 (−0.92–0.62)	−0.29 (−1.05–0.47)	0.09
	Model 2	−0.46 (−0.98–0.05)	0.08	0.00	0.37 (−0.53–1.26)	0.05 (−0.74–0.84)	−0.18 (−0.96–0.60)	−0.31 (−1.08–0.45)	0.08

Model 1: adjusted for body mass index, total energy intake, diet quality, physical activity, and follow-up time. Model 2: model 1 additionally adjusted for education level and smoking status. ^a^ Betas (95%CI) for intraocular pressure (IOP) by total dietary nitrate intake, nitrate intake from vegetables, and nitrate intake from non-vegetable food sources (as continuous variables) analyzed using linear regression. ^b^ Test for trend conducted using median value for each quintile (total dietary nitrate intake: quintile 1 = 48.8 mg/day; quintile 2 = 69.0 mg/day; quintile 3 = 86.4 mg/day; quintile 4 = 114.0 mg/day; quintile 5 = 213.0 mg/day; nitrate intake from vegetables: quintile 1 = 34.6 mg/day; quintile 2 = 54.2 mg/day; quintile 3 = 72.0 mg/day; quintile 4 = 98.1 mg/day; quintile 5 = 196.8 mg/day; nitrate intake from non-vegetable food sources: quintile 1 = 10.1 mg/day; quintile 2 = 11.9 mg/day; quintile 3 = 14.1 mg/day; quintile 4 = 15.8 mg/day; quintile 5 = 21.1 mg/day).

## Data Availability

Data can be obtained upon request. Requests should be directed towards the management team of the Rotterdam Study (datamanagement.ergo@erasmusmc.nl), which has a protocol for approving data requests. Due to restrictions based on privacy regulations and informed consent of the participants, data cannot be made freely available in a public repository.

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
