# Peer review of "Dietary Nitrate Intake Is Associated with Decreased Incidence of Open-Angle Glaucoma: The Rotterdam Study"

_nutrients, 2022, doi:10.3390/nu14122490_

Round 1

Reviewer 1 Report

What was the definition of visual field loss that defined glaucoma? 

Was glaucoma also defined by an objective finding, such as the optic disc finding or imaging of the optic disc?

Were the incident cases of glaucoma or controls being treated in any case prior to their diagnoses?

"As we did not observe an association between dietary nitrate intake and IOP, nitrate may protect against glaucoma via other, IOP-independent, mechanisms." The authors should avoid making statements such as "nitrate is protective" since what they have found is a statistical association.

There is very little association between ischemia--reperfusion and glaucoma pathogenesis hence the statements on page nine are irrelevant.

If I am calculating correctly it has been over 20 years since the incident cases average followup occurred, that is 10-20 years after the start of a study in 1991.  How much might the diet of persons in this population have changed since then?  Thus, is this study still relevant to today? 30 years ago eating more vegetables was not the vogue it is today.

This study would have been that much stronger if a follow-up food questionnaire had been administered.  The long space between the questionnaire and the incidence of glaucoma (only mentioned at the last part of the limitations) is quite a major limitation.

Past studies of the Rotterdam population mention several other risk factors which were not taken into account in this study including myopia, cup-to-disc ratio, and family history of glaucoma, as well as other features such as exfoliation. How big is the nitrate association compared to these?

Overall the finding seems to be an association without a rationale since in effect of nitric oxide presumably related to past nitrate intake would, by their own discussion, change eye pressure or blood pressure yet neither variable was associated with the incident glaucoma.

Reviewer 2 Report

The authors have conducted a case-control analysis based on a large epidemiological prospective study and investigated an association between dietary nitrate intake and incidence of OAG.

The study has been well planned and performed. Chapter Material and Methods contain all necessary information. The writing is of perfect quality and the results are clearly presented.

The results and conclusions of this study are important.

I would only like to ask about one fact that I find intriguing. The authors have found that participants without OAG had a significantly higher BMI than those with OAG. Their mean BMI was found to be 27.1 kg/m2 which is considered overweight. Lower risk of OAG would be related to a higher total dietary nitrate intake. “Previous research has shown that a higher dietary nitrate intake was associated with significantly wider retinal arterioles.[46] Widening of retinal arteriolar caliber is not only associated with lower risk of cardiovascular and cerebrovascular diseases [47,48], but also with lower risk of glaucoma”. On the other hand, “compared with individuals with a normal BMI (defined as a BMI of 18.5 to 24.9), lifetime risks for incident CVD [cardiovascular disease] were higher in middle-aged adults in the overweight and obese groups” Khan SS, Ning H, Wilkins JT, et al. Association of Body Mass Index With Lifetime Risk of Cardiovascular Disease and Compression of Morbidity. JAMA Cardiol. 2018;3(4):280–287. doi:10.1001/jamacardio.2018.0022

I wonder if the authors could comment on this in the Discussion.
